# Complete Plastome of *Physalis angulata* var. *villosa*, Gene Organization, Comparative Genomics and Phylogenetic Relationships among Solanaceae

**DOI:** 10.3390/genes13122291

**Published:** 2022-12-05

**Authors:** Xiaori Zhan, Zhenhao Zhang, Yong Zhang, Yadi Gao, Yanyun Jin, Chenjia Shen, Huizhong Wang, Shangguo Feng

**Affiliations:** 1College of Life and Environmental Science, Hangzhou Normal University, Hangzhou 310036, China; 2Zhejiang Provincial Key Laboratory for Genetic Improvement and Quality Control of Medicinal Plants, Hangzhou Normal University, Hangzhou 310036, China; 3Quzhou Academy of Agriculture and Forestry Sciences, Quzhou Municipal Bureau of Agriculture and Rural Affairs, Quzhou 324000, China

**Keywords:** *Physalis angulata* var. *villosa*, plastome, repeat analysis, SSRs, comparative genomics, phylogenetic relationship

## Abstract

*Physalis angulata* var. *villosa*, rich in withanolides, has been used as a traditional Chinese medicine for many years. To date, few extensive molecular studies of this plant have been conducted. In the present study, the plastome of *P. angulata* var. *villosa* was sequenced, characterized and compared with that of other *Physalis* species, and a phylogenetic analysis was conducted in the family Solanaceae. The plastome of *P. angulata* var. *villosa* was 156,898 bp in length with a GC content of 37.52%, and exhibited a quadripartite structure typical of land plants, consisting of a large single-copy (LSC, 87,108 bp) region, a small single-copy (SSC, 18,462 bp) region and a pair of inverted repeats (IR: IRA and IRB, 25,664 bp each). The plastome contained 131 genes, of which 114 were unique and 17 were duplicated in IR regions. The genome consisted of 85 protein-coding genes, eight rRNA genes and 38 tRNA genes. A total of 38 long, repeat sequences of three types were identified in the plastome, of which forward repeats had the highest frequency. Simple sequence repeats (SSRs) analysis revealed a total of 57 SSRs, of which the T mononucleotide constituted the majority, with most of SSRs being located in the intergenic spacer regions. Comparative genomic analysis among nine *Physalis* species revealed that the single-copy regions were less conserved than the pair of inverted repeats, with most of the variation being found in the intergenic spacer regions rather than in the coding regions. Phylogenetic analysis indicated a close relationship between *Physalis* and *Withania*. In addition, *Iochroma, Dunalia, Saracha* and *Eriolarynx* were paraphyletic, and clustered together in the phylogenetic tree. Our study published the first sequence and assembly of the plastome of *P. angulata* var. *villosa*, reported its basic resources for evolutionary studies and provided an important tool for evaluating the phylogenetic relationship within the family Solanaceae.

## 1. Introduction

*Physalis angulata* L. var. *villosa* Bonati belongs to the family Solanaceae and has important potential pharmaceutical value [1,2]. It is distributed mainly in Vietnam and the Hubei, Jiangxi and Yunnan Provinces of China [3]. It is interesting to note that, unlike other *Physalis* plants, *P. angulata* var. *villosa* has a remarkable characteristic: its whole body is covered with short white fluff (Figure 1) [1]. Additionally, more importantly, *P. angulata* var. *villosa* has been widely used for various traditional medicinal treatments in China. *P. angulata* var. *villosa* is rich not only in minerals and antioxidants, but also in a range of pharmacologically-active withanolides, including physagulins A–Q, withangulatins A−I, physangulidines A−C, physalins B, D, F, G, and H, withaminimin, physangulatins A−N, and withaphysalins Y and Z, and many of them have anti-inflammatory, anti-bacterial or anti-cancer activities [2,4]. *P. angulata* var. *villosa* also has important edible and ornamental values, so it is much appreciated by people in China [1,5].

Although studies into several pharmacologically active ingredients and their functions have been carried out in *P. angulata* var. *villosa* over the past few decades, no exclusive genetic or genomic studies of this plant have been conducted to date. Plastomes (or named chloroplast genomes) play an important role in carbon fixation and stress response in plants, and the plastome is an important organelle in plant cells [6]. The plastomes of most land plants have typical, highly conserved four-part structures: one large single-copy (LSC) region, one small single-copy (SSC) region and a pair of inverted repeats (IR), with the plastome size ranging from 120 to 170 kb [7,8]. Compared with nuclear DNA sequences, the gene content and gene composition of plastomes are relatively conservative, and their evolutionary rate is relatively slow [9,10]. In addition, many studies have observed expansion, contraction, and reversal or gene rearrangement events in the plastomes of angiosperms, which may be the result of differential rates of insertions/deletions (indels) and substitutions during plant evolution [11,12,13,14]. With the rapid development of next-generation sequencing technology, more and more plant plastomes have been sequenced and have been widely used in plant phylogeny, taxonomy and identification [15,16,17,18,19].

In recent years, the identification and systematic classification of *Physalis* species and the phylogenetic relationships within Solanaceae have become a focus of attention for researchers [1,20,21]. Olmstead et al. (2008) studied the phylogenic relationships of Solanaceae based on the chloroplast DNA regions *ndhF* and *trnLF*, the results of which not only provided a framework phylogeny of Solanaceae, but also reveal some problems that still require further study [21]. For example, the study showed that genera appear to be non-monophyletic, including *Lycianthes*, *Lycium* and *Physalis*, etc. [21]. Prior to the current study, universal DNA barcoding sequences, such as the plastid *psbA*–*trnH* intergenic spacer region [22], nuclear internal transcribed spacers (ITSs) [1,20] and some traditional molecular markers, such as simple sequence repeats (SSRs) [23] and inter simple sequence repeats (ISSRs) [24,25], have been used to identify species and to construct phylogenies of *Physalis* species. However, specific nuclear or plastid gene sequences are relatively short, providing only limited genetic information, and have proved unsuccessful for distinguishing very closely related species [6]. Thus, the phylogenetic trees constructed from such sequences need more advanced techniques and methods for verification. By contrast, most plant plastomes can provide significant quantities of genetic information at the genome-wide evolutionary level and can improve the identification of plant species and better elucidate the relationship between plant taxonomy and phylogeny [26,27,28]. The abundance of plastome sequencing data has accelerated the progress of research into the relationship between plant evolution and phylogeny [6,29]. Increasingly, plastome sequences have been used by researchers to study phylogenetic relationships at different taxonomic levels [27,30,31,32].

In the present study, we sequenced and characterized the plastome of *P. angulata* var. *villosa* and compared the genome with plastomes from other solanaceous species. The purpose of our study was to examine both the phylogenetic relationships and the plastome evolution of *P. angulata* var. *villosa* within the Solanaceae. In addition, we also identified SSRs, repeat sequences and the hotspots of sequence variation in *Physalis* species, which might provide some important information for the development of polymorphic molecular markers for germplasm identification and evolutionary studies of *Physalis*.

## 2. Materials and Methods

### 2.1. Plant Material

*P. angulata* var. *villosa* plants were collected from Hangzhou (30°20′84′′ N, 120°21′20.1′′ E) in Zhejiang Province, China and their identity was confirmed by Prof. Huizhong Wang (Hangzhou Normal University). The voucher sample was deposited in the Zhejiang Provincial Key Laboratory for Genetic Improvement and Quality Control of Medicinal Plants, Hangzhou Normal University, China (Voucher specimen PHZ002). No permission was necessary for the collection of *P. angulata* var. *villosa*, which is widely distributed in China and is not listed among the national key protected plants.

### 2.2. DNA Extraction and PCR Amplification

Total genomic DNA was extracted from 0.15 g of fresh leaf material (pooled equal weights of leaves from six individual *P. angulata* var. *villosa* plants) using the Plant Genomic DNA Extraction Kit (Shanghai Sangon Biological Engineering Technology and Service Co., Ltd., Shanghai, China), according to the manufacturer’s protocol.

### 2.3. Library Construction, Sequencing and Assembly

The genomic DNA was sequenced using the Illumina HiSeq 2000 platform. The sequencing work was carried out at the Germplasm Bank of Wild Species in Southwest China, Kunming Institution of Botany, Chinese Academy of Sciences, Kunming, China. Raw reads were filtered using the NGS QC Toolkit v. 2.3.3 [33] with a cut-off value of 80 for the percentage of read lengths and a cut-off value of 30 for PHRED quality scores. The plastome assembly, using the high-quality clean reads, was conducted using CLC genomics workbench 8 [34] and SOAPdenovo [35] with a *k* mer of 63 and a minimum overlap length of 1 kb. The plastome of *Physalis peruviana* (GenBank accession number: MH019242) was used as the reference sequence for genome assembly.

### 2.4. Genome Annotation and Sequence Analysis

The online plastid genome annotation program Dual Organellar GenoMe Annotator (DOGMA) [36] was used to annotate the genes in the *P. angulata* var. *villosa* plastome. The positions of the start and stop codons were checked manually. The tRNAscan-SE v2.0 [37] software was used to confirm the tRNA genes. Organellar Genome Draw (ORGDRAW) [38] was used to draw the map of the plastome. The complete plastome sequence of *P. angulata* var. *villosa* was submitted to GenBank (Accession number OM257167). MEGA7 [39] was used to analyze the relative synonymous codon usage (RSCU), codon usage and base composition. The RNA editing sites in the plastome of *P. angulata* var. *villosa* was predicted using the PREP suite with a cut-off value of 8.0 [40].

### 2.5. Repeat Sequence Analysis

Long repeat sequences were detected using the online program REPuter [41] with the following parameters: cut-off n ≥ 30 bp, a Hamming distance of 3, and a 90% similarity among repeat units. The Perl script MIcroSAtellite (MISA) program [42] was used to detect SSRs with the following settings: ten repeat units for mononucleotide, five for dinucleotide, four for trinucleotide, and three for tetra-, penta-, or hexanucleotide SSR motifs.

### 2.6. Genome Comparison

The online software mVISTA was used to compare the variation in the complete plastomes of nine *Physalis* species, using the annotation of *P. angulata* var. *villosa* as a reference in the global alignment Shuffle-LAGAN mode [43]. The structure variation in SC/IRs borders between nine *Physalis* species were visualized by using the R script of IRscope software (https://irscope.shinyapps.io/irapp/ (accessed on 25 November 2022).

### 2.7. Phylogenetic Analysis

For phylogenetic analysis, 80 plastomes of Solanaceae species, representing 23 genera deposited in GenBank, were downloaded (Appendix A). The plastomes of four non-solanaceous species, namely *Bacopa monnieri*, *Digitalis lanata*, *Rehmannia glutinosa*, and *Scrophularia buergeriana*, were also downloaded and designated as outgroups (Appendix A). MAFFT version 7 [44] was used for the alignment of these selected, complete, solanaceous plastome sequences. Maximum likelihood (ML) and neighbor-joining (NJ) phylogenetic analyses were performed in MEGA7 [39]. The most suitable models were selected after model testing in MEGA7, and the general time reversible (GTR+G+I) substitution model for evolution was used with 1000 bootstrap repeats to ascertain branch support.

## 3. Results

### 3.1. Chloroplast Assembly and Genome Feature

The complete plastome of *P. angulata* var. *villosa* was 156,898 bp in size, with a typical quadripartite structure containing a large single-copy region (LSC, 87,108 bp), a small single-copy region (SSC, 18,462 bp), and a pair of inverted repeats (IRA and IRB, 25,664 bp for each) (Figure 2 and Table 1). GC content in the entire *P. angulata* var. *villosa* plastome was 37.52%, whereas the LSC, SSC and IR regions had GC values of 35.58%, 31.33% and 43.05%, respectively (Table 1). The results suggested that IR regions had higher GC contents than did the LSC and SSC regions (Table 1).

The complete plastome *P. angulata* var. *villosa* contains 131 genes, including 114 different genes and 17 genes duplicated in IR regions. The number of protein-coding genes, rRNA genes and tRNA genes in the *P. angulata* var. *villosa* plastome is 85, eight and 38, respectively (Figure 2, Table 1 and Table 2). Four rRNA, seven tRNA and six protein-coding genes are duplicated in the two copies of the IR regions, whereas 61 protein-coding genes and 23 tRNA genes are in the LSC region, with the remaining 11 protein-coding genes and one tRNA gene being in the SSC region. There are 19 different genes containing introns (Table 3), including seven tRNA genes and 12 protein-coding genes, six of which (*ndhB*, *rpl2*, *trnA-UGC*, *trnE-UUC*, *trnI-GAU* and *trnV-UAC*) are located in the pair of IR regions, whereas only one intron-containing gene (*ndhA*) is located in the SSC region, with the other 11 being in the LSC region. In addition to that, the *rps12* gene is divided into 5′-*rps12* in the LSC region and 3′-*rps12* in the pair of IR regions. Two genes, namely *clpP* and *ycf3*, have two introns each, whereas the other 16 genes have one intron. The *trnK-UUU* gene has the largest intron (2509 bp in length), with the *matK* gene harbored within it (Table 2 and Table 3).

The codon usage and codon-anticodon recognition patterns of the plastome of *P. angulata* var. *villosa* were analyzed using the nucleotide sequences of protein-coding genes and tRNA genes (Table 4). The genes in the *P. angulata* var. *villosa* plastome are encoded by 52,299 codons. The frequency rates of the individual amino acids encoded in the *P. angulata* var. *villosa* plastome range from 1.25% to 10.83% (Figure 3). The codons that code for the amino acid leucine are the most frequent, with 5144 of the total codons (10.83%), followed by arginine (4558 codons, 9.60%), isoleucine (4399 codons, 9.26%), phenylalanine (3721 codons, 7.84%), serine (3585 codons, 7.55%) and lysine (3119 codons, 6.57%) having relatively high usage rates, while the codons that code for the amino acid glycine appeared to be the least frequent, at 594 codons (1.25%). The relative synonymous codon usage (RSCU) values ranged from 0.41 (CGC) to 1.94 (AGA) (Table 4). Almost all A- or U-ending codons (28 codons) had RSCU values >1, except for CGU, CUA, AGU and AUA (RSCU = 0.73, 0.85, 0.86 and 0.97, respectively), whereas almost all G- or C-ending codons had RSCU values < 1 except for ACC, CCC, UCC and UUG (1.03, 1.04, 1.15 and 1.26, respectively). The amino acids tryptophan and methionine did not exhibit codon bias because they had RSCU values = 1.

The RNA editing sites present in the *P. angulata* var. *villosa* plastome were predicted (Appendix A). The highest number of conversions in the codon positions are from proline to leucine (28 sites), followed by proline to serine (23 sites), threonine to isoleucine (21 sites), histidine to tyrosine (20 sites) and serine to leucine (20 sites), whereas proline to leucine and arginine to tryptophan have the lowest number of conversions, with only one site each (Appendix A). The total number of editing sites observed in the *P. angulata* var. *villosa* plastome is 150, distributed between 25 of the protein-coding genes. The gene with the highest number of editing sites is *psaB* (18 sites), followed by *rpoB* (14 sites), *ndhB* (13 sites), *rpoC1* (11 sites), *rpoC2* (11 sites) and *ndhF* (ten sites). In contrast, the *petB*, *ndhG*, *psbE*, *psbF* and *rpl20* genes have the lowest number of editing sites, with only one editing site each.

### 3.2. Repeat Element Analysis

Repeat element sequences in the *P. angulata* var. *villosa* plastome were analyzed, and the results are shown in Table 5. In total, three types of repeats, namely forward, palindromic and reverse repeats, were detected, except for the complementary repeats (Table 5). The results showed that a total of 38 repeats, namely 22 forward repeats, 14 palindromic repeats and two reverse repeats, was identified in the *P. angulata* var. *villosa* plastome (Table 5). Most of the repeats (84.21%), including 19 forward, 11 palindromic and two reverse repeats, had sizes of 30–39 bp, followed by 13.16% of the repeats being 40–49 bp (including three forward and two palindromic repeats), whereas 50–59 bp repeats were the least frequent (2.63%), including only one palindromic repeat. Most (55.26%) of the repeats were distributed in the intergenic spacers (IGS), but only two repeats (5.26%) were located in tRNA genes. The other 15 repeats were found in the protein-coding genes *psbT*, *psaB*, *ndhA*, *ycf1*, *ycf2*, and *ycf3*. The repeats in the plastomes of the eight other *Physalis* species were also analyzed (Figure 4). Compared with *P. angulata* var. *villosa*, 42, 43, 42, 39, 44, 34, 40 and 35 repeats were detected in the *Physalis angulata*, *Physalis minima*, *Physalis pubescens*, *Physalis peruviana*, *Physalis chenopodifolia*, *Physalis philadelphica*, *Physalis pruinosa* and *Physalis alkekengi* var. *franchetii* (synonyms for *Alkekengi officinarum*) plastomes, respectively (Figure 4A). Although the number and length of the repeats differed among the nine *Physalis* species, most of the repeats in the nine species were distributed in the length range 30–39 bp (Figure 4B).

### 3.3. SSR Analysis

SSRs are a very important class of molecular marker, which are widely distributed in plastome. In the current study, a total number of 57 SSRs was detected in the plastome of *P. angulata* var. *villosa* (Figure 5A). Most of the SSRs were mononucleotide (39, 68.42%), followed by seven tetranucleotide SSRs (12.28%), six dinucleotides SSRs (10.53%) and five trinucleotides SSRs (8.77%). Among them, the mononucleotide SSRs were composed of A, T or C, of which T constituted the majority (61.54%). Most of the SSRs were detected in the intergenic spacer (IGS) regions (68.42%), followed by 29.82% of SSRs detected in the protein-coding regions (Figure 5B), whereas SSRs located in the tRNA genes showed the lowest frequency (1.75%). The results also showed that most SSRs were in the LSC region (71.93%), compared with the IR (14.04%) and the SSC regions (8.77%) (Figure 5C).

### 3.4. Comparative Genomics Analysis

The plastome sequences of nine *Physalis* species were compared, including *P. angulata* var. *villosa* (156,898 bp), *P. angulata* (156,905 bp), *P. minima* (156,692 bp), *P. pubescens* (157,007 bp), *P. peruviana* (156,706 bp), *P. chenopodifolia* (156,888 bp), *P. philadelphica* (156,804 bp), *P. pruinosa* (156,706 bp) and *P. alkekengi* var. *franchetii* (156,578 bp). To investigate the level of plastome sequences variation between *P. angulata* var. *villosa* and the eight other *Physalis* species, mVISTA software was used to align the sequences, with the annotation of *P. angulata* var. *villosa* being used as the reference (Figure 6). The results showed that the *Physalis* plastomes were highly conserved, but some level of variation was detected. Compared with the LSC and SSC regions, the pair of IR regions showed low levels of variation. Furthermore, the protein-coding genes showed less divergence than did the non-coding regions, especially the intergenic spacer regions. The intergenic spacer regions with a high variation level included *petN*–*psbM*, *trnL-UAA*–*trnF-GAA*, *ndhC*–*trnV-UAC*, *rbcL*–*accD*, *accD*–*psbI*, *petA*–*psbJ*, *trnL-UAG*–*ccsA*, *trnQ-UUG*–*psbK*, *atpH*–*atpI* and *rpl32*–*trnL*-*UAG*. In addition to that, protein-coding genes, such as *ycf1*, *ycf2*, *ndhF*, *rps19* and *ccsA*, also showed high sequence variation.

The main factors determining plastome size variation are the expansion and contraction of IR regions, which play an important role in plant evolutionary history. In our study, the sizes and the boundaries of the LSC, SSC, and IR regions of the nine *Physalis* plastomes were compared (Figure 7). The lengths of the IR regions ranged from 24,777 bp to 25,685 bp between the nine *Physalis* plastomes, indicating some potential expansion and contraction had occurred in the IR regions, which might be the main reasons for the differences in genome length of *Physalis* plastomes. The *rps19*, *rpl2*, *rpl23* and *trnH* genes were located near the LSC/IR border of the nine *Physalis* plastomes and exhibited slight variation in the number of nucleotides, whereas the *ycf1* and *ndhF* genes were found near the IR/SSC border. The *rps19* gene crossed the LSC/IRB regions in the six *Physalis* plastomes (*P. angulata* var. *villosa*, *P. minima*, *P. pubescens*, *P. chenopodifolia*, *P. philadelphica* and *P. alkekengi* var. *franchetii*), but it was located in the LSC region near the LSC/IRB border in *P. angulata*, and at the IRA/LSC border in only the plastome of *P. minima*. The *rpl2* gene was located at the IRA/LSC boundary in *P. angulata* var. *villosa*, *P. angulata*, *P. minima*, *P. pubescens* and *P. alkekengi* var. *franchetii*, but was absent from the IRA/LSC boundary in *P. peruviana*, *P. chenopodifolia*, *P. philadelphica* and *P. pruinosa*. Notably, the *rpl2* gene crossed the LSC/IRB borders and extended to the LSC region in *P. peruviana* and *P. pruinosa*, which is different from the other seven species.

### 3.5. Phylogenetic Analysis

To determine the phylogenetic relationship and tribal positions among *P. angulata* var. *villosa* and the eight other *Physalis* species within the Solanaceae, we used the plastomes of 80 Solanaceae species, representing 23 genera of seven tribes (Physaleae, Capsiceae, Solaneae, Datureae, Lycieae, Hyoscyameae and Nicotianeae) from three subfamilies (Solanoideae, Nicotianoideae and Petunioideae), to construct phylogenetic trees. The two phylogenetic analyses were performed using the maximum likelihood (ML) or the neighbor-joining (NJ) methods, with *Bacopa monnieri*, *Digitalis lanata*, *Rehmannia glutinosa* and *Scrophularia buergeriana* as outgroups. The phylogenetic results from the ML and NJ analyses were similar and divided all the species into seven groups with very high support (Bootstrap (BS) = 100%) (Figure 8 and Appendix A).

As described in Figure 8 and Appendix A, group I was the most complex group, comprising 29 species of eight genera from tribe Physaleae of subfamily Solanoideae, which could be further divided into three subgroups (I-1, I-2 and I-3). Subgroup I-1 contained 17 species: three species from genus *Dunalia*, ten species from genus *Iochroma* and one each from genera *Acnistus*, *Eriolarynx*, *Saracha* and *Vassobia*. Subgroup I-2 contained three *Withania* species: *Withania riebeckii*, *Withania adpressa* and *Withania coagulans*. Nevertheless, we found that *P. angulata* var. *villosa* and the eight other *Physalis* species were closely related to these genera and were grouped separately into subgroup I-3. Group II contained all 10 species from the genus *Capsicum* of tribe Capsiceae of subfamily Solanoideae. Group III contained all the 13 *Solanum* species from tribe Solaneae and one species from tribe Physaleae (*Tubocapsicum anomalum*). Two species, *Datura stramonium* and *Trompettia cardenasianum* from tribe Datureae of subfamily Solanoideae, were grouped into group IV. Group V included 13 species from eight genera, and was further divided into two subgroups, V-1 and V-2. Subgroup V-1 contained four *Lycium* species from tirbe Lycieae of subfamily Solanoideae, whereas subgroup V-2 contained two *Anisodus* species (*Anisodus tanguticus* and *Anisodus acutangulus*), two *Physochlaina* species (*Physochlaina orientalis* and *Physochlaina physaloides*), and one species each from the genera *Atropa*, *Atropanthe*, *Hyoscyamus*, *Przewalskia* and *Scopolia* within tribe Hyoscyameae of subfamily Solanoideae. All 10 species from genus *Nicotiana* of tribe Nicotianeae of subfamily Nicotianoideae were classified in group VI, whereas the two *Petunia* species from subfamily Petunioideae analyzed were distant from any other Solanaceae species tested and were assigned into group VII.

## 4. Discussion

In the present study, we sequenced the plastome of *P. angulata* var. *villosa* using Illumina sequencing technology and compared it with the published plastomes of the other eight *Physalis* species. The analysis of the plastome showed that the plastome of *P. angulata* var. *villosa* had a typical quadripartite structure, which contained a pair of IR regions (IRA and IRB), one SSC region and one LSC region. The organization and structure of the *P. angulata* var. *villosa* plastome was similar to that of the other *Physalis* plastomes [45]. The size of the *P. angulata* var. *villosa* plastome (156,898 bp) was comparable to that of other sequenced plastomes of members of the Solanaceae, being longer than those of *Datura stramonium* [46], *Solanum brevicaule* [47] and *Withania somnifera* [48], but shorter than those of *Iochroma ellipticum* (GenBank accession: KU323367), *P. pubescens* [45] and *Eriolarynx fasciculata* (GenBank accession: KU306938). The GC content of the *P. angulata* var. *villosa* plastome was 37.52%, which was similar to that of many other Solanaceae species [45,49,50,51]. In addition, the *rps12* gene in the *P. angulata* var. *villosa* plastome was found to be a *trans*-spliced gene, as had been reported in other species [48,52]. Although the length of the *P. angulata* var. *villosa* plastome was different from that of other Solanaceae species, the arrangement and gene contents of the plastomes were similar [45,53,54].

Repeat elements are widely present in plant plastomes and are associated with recombination and rearrangement events [10,55,56]. Furthermore, these repeat sequences are the basis of population and phylogenetic studies [28,57]. A total of 38 repeats were identified in the *P. angulata* var. *villosa* plastome, which was more than of the corresponding number in *Withania somnifera* [48], but significantly fewer than that in *Physalis chenopodifolia* [53]. The lengths of 84.21% of the repeat elements in the *P. angulata* var. *villosa* plastome was between 30 and 39 bp, a finding similar to that of most other Solanaceae plastomes [45,48,54]. Interestingly, although *P. angulata* var. *villosa* and *P. angulata* are genetically closely related, we found some differences in repeat sequences within the number and length between these two species, which might help us understand their phylogenetic differences. Chloroplast SSRs (cpSSRs) are short repeats distributed in plastomes and inherited from a single parent and are often used as DNA molecular makers in genetic diversity, species identification and phylogenetic studies [58,59,60]. CpSSR analysis revealed a total of 57 SSRs in the *P. angulata* var. *villosa* plastome, most of which were mononucleotides, primarily A and T, a finding similar to that reported in many other plants [45,61,62]. Most (77.19%) of the cpSSRs of *P. angulata* var. *villosa* were in the LSC region, whereas the number of cpSSRs distributed in the SSC region was the lowest (5.26%). The cpSSRs detected in the present study will be useful in genetic diversity and population structure studies of *P. angulata* var. *villosa*, as well as with respect to genetic relationship and species identification investigations of the genus *Physalis*.

The variation in plastome size is mainly due to the expansion and contraction of IR regions [63]. After comparing the plastomes among the nine *Physalis* species in the current study, contraction and expansion of the IR regions were detected in the *P. angulata* var. *villosa* plastome and other sequenced *Physalis* plastomes. The boundary regions between the SSC and the two IR regions were relatively highly conserved, and the distribution and location of genes in these regions were consistent. However, the boundary regions between the LSC and the two IR regions varied greatly, which were similar with the finding in many angiosperm plants [64,65]. The contraction and expansion of IR borders can reflect the genetic relationship of plant species [66,67]. Despite the similar size of the IR regions between *P. angulata* var. *villosa* and the other *Physalis* species, some level of expansion and contraction was detected. There were variations in the border of the LSC and IR regions among the nine *Physalis* species, mainly based on the position of *rps19*, *rpl2* and *trnH*. In *P. angulata* var. *villosa*, *P. angulata*, *P. pubescens* and *P. alkekengi* var. *franchetii*, one *rpl2* gene was located in the IRA region, whereas only one *rps19* gene spanned the LSC/IRB boundary, with most of the sequence being present in the LSC region. In *P. minima*, one of the duplicated *rps19* copies spanned the LSC/IRB boundary, whereas the other was located in the IRA region. In *P. peruviana*, *P. chenopodifolia*, *P. pruinosa* and *P. philadelphica*, no *rpl2* genes were found in the IRA/LSC border. It was observed that there were extensions of the IR region into the LSC regions for *P. peruviana* and *P. pruinosa*, resulting in these two species having relatively long LSC regions with 88,718 bp and 88,758 bp, respectively.

The result of the comparative genomic analysis using mVISTA showed that the genomes of *P. angulata* var. *villosa* and other eight *Physalis* species were relatively highly conserved, with a low degree of sequence divergence. Any variations mainly occurred in the non-coding regions of the plastomes due to the results of insertion and deletion, a finding which is consistent with most angiosperm plants [11,64]. The comparative genome analysis results also revealed some variable regions in the tested *Physalis* species, namely *petN*–*psbM*, *trnL-UAA*–*trnF-GAA*, *ndhC*–*trnV-UAC*, *rbcL*–*accD*, *accD*–*psbI*, *petA*–*psbJ*, *trnL-UAG*–*ccsA*, *trnQ-UUG*–*psbK*, *atpH*–*atpI*, *trnL*-*UAG*, *ycf1*, *ycf2*, *ndhF*, *rps19* and *ccsA*, which could be used as potential DNA barcodes for the identification of *Physalis* species, as well as for resolving phylogenetic relationships in the family Solanaceae.

Many previous studies have indicated that the phylogenetic classification of the genus *Physalis* is complex, due to the large number of species, wide distribution, and relatively similar morphological characteristics [1,20,21]. In the present study, phylogenetic trees were constructed from the plastomes of 80 species representing 23 genera of seven tribes from three subfamily of the family Solanaceae, using NJ and ML methods. The two phylogenetic trees showed similar topology with a very high support rate of 100% for all the groups. The results revealed that all the species in the genus *Physalis* were grouped into a separate subgroup, with the genetic relationship between *P. angulata* var. *villosa* and *P. angulata* being the closest, whereas that between *P. angulata* var. *villosa* and *P. alkekengi* var. *franchetii* was the most distant. The finding not only corresponded to current taxonomy of genus *Physalis*, but also further confirmed the relatively distant genetic relationship between *P. alkekengi* var. *franchetii* and other *Physalis* species [1,20]. The results also showed that the tested *Physalis* species have relatively close genetic relationships with the solanaceous genera *Withania*, *Iochroma*, *Dunalia*, *Saracha*, *Eriolarynx*, *Vassobia*, and *Acnistus*, whereas the *Physalis* species studied were most distantly related to the *Petunia* species tested from subfamily Petunioideae, which confirms some previous studies and improves the phylogenetic map of the Solanaceae [1,20,21,45]. The clustering results showed that most species from the same genus of the family Solanaceae were monophyletic, but interestingly, a few species of some genera were paraphyletic in the phylogenetic evolution of the Solanaceae. For example, the genetic relationships among *Iochroma*, *Dunalia*, *Saracha* and *Eriolarynx* species were complex, confirming earlier findings [20,21,45]. In addition, interestingly, our study also showed that at the tribe level, all species from the same tribe tended to cluster together, except *Tubocapsicum anomalum* from tribe Physaleae. We speculated that the reason why *T. anomalum* were distant from the other species of tribe Physaleae might be that the plastome sequence of this species was downloaded from the GenBank database and there was only one sequence; it was debatable whether the sequence information and species were consistent due to the similar morphological characteristics of some Solanaceae species. Therefore, the phylogenetic position of this species may need to be further verified by more plastome sequences.

## 5. Conclusions

In the present study, we sequenced and reported the complete plastome of *P. angulata* var. *villosa*, providing new, valuable plastid genomic resources for the genus *Physalis*. The plastome of *P. angulata* var. *villosa* has a typical angiosperm plastome structure and gene content and is comparable to other *Physalis* plastomes. The repeat sequences and SSRs identified in this study could be used as valuable tools for evolutionary research within the genus *Physalis* in the future. The comparative genomic analyses of nine *Physalis* species showed that some variable hotspots could be used to develop DNA barcodes for the identification of the species. The present study also revealed the taxonomic position and genetic relationships of major genera in the Solanaceae, showing that several species of some genera were paraphyletic during phylogenetic evolution.

## Figures and Tables

**Figure 1 genes-13-02291-f001:**
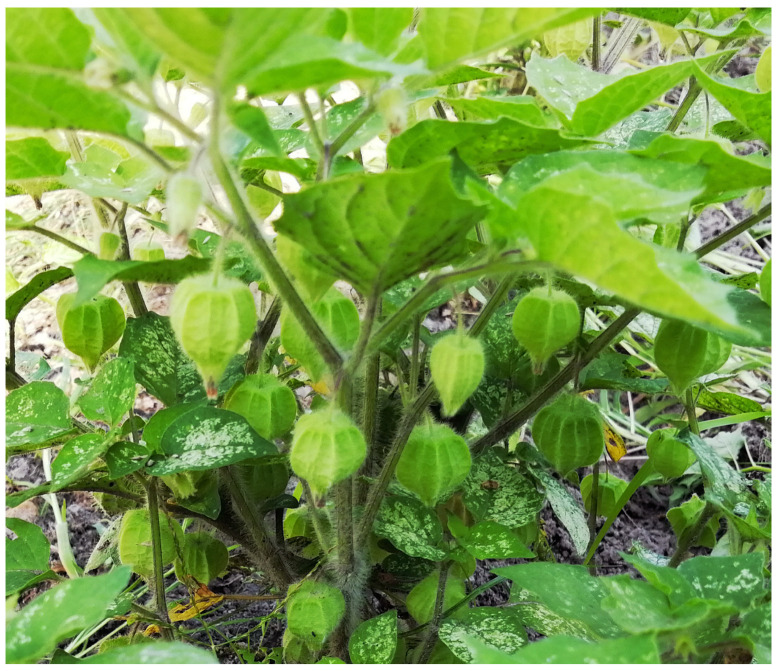
Plant morphology of *P. angulata* var. *villosa* in natural habitats.

**Figure 2 genes-13-02291-f002:**
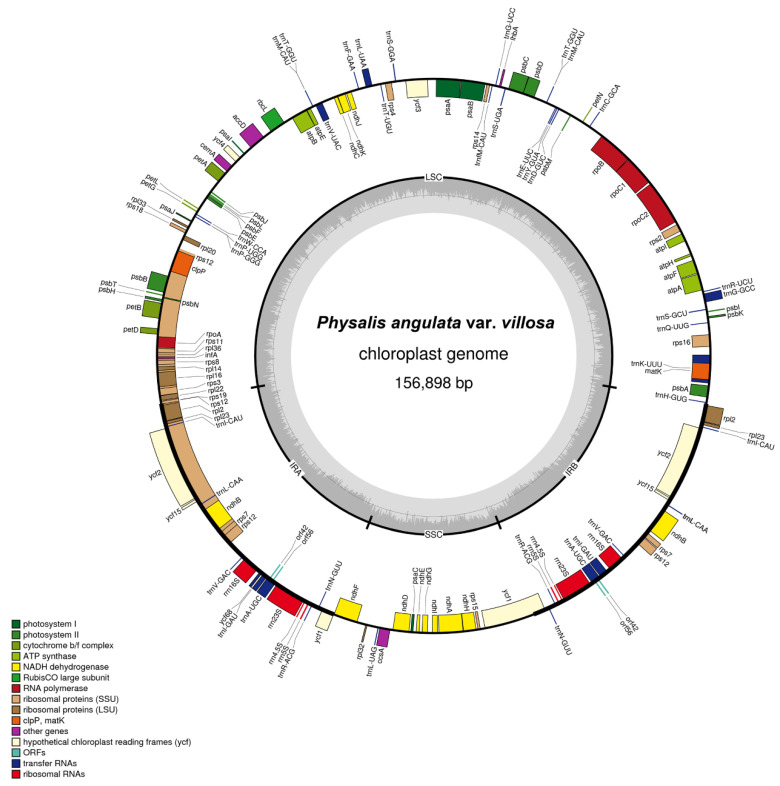
Gene map of the plastome of *P. angulata* var. *villosa*. Genes outside the circles are transcribed in a counterclockwise direction, and genes inside the circles in a clockwise direction. Known functional genes are color coded. AT and GC contents are denoted by the light and dark grays in the inner circle, respectively. LSC indicates large single-copy region and SSC indicates small single-copy region, whereas IRA and IRB indicate inverted repeats.

**Figure 3 genes-13-02291-f003:**
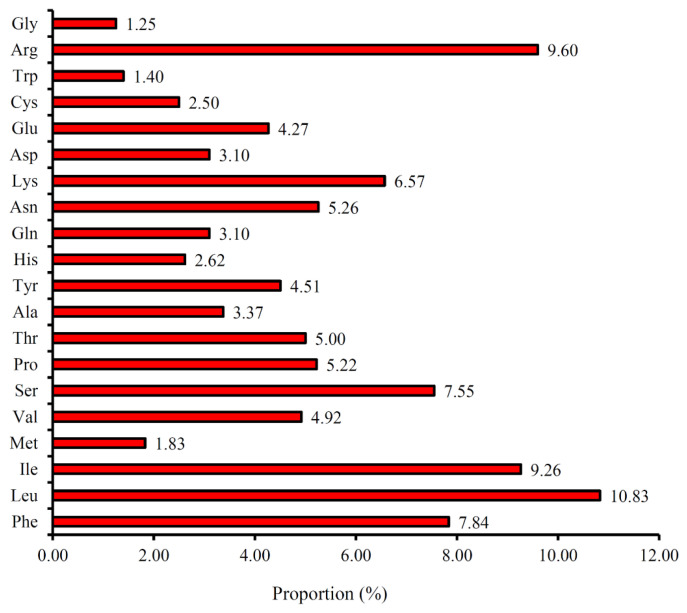
Amino acid frequencies in the *P. angulata* var. *villosa* plastome protein-coding sequences.

**Figure 4 genes-13-02291-f004:**
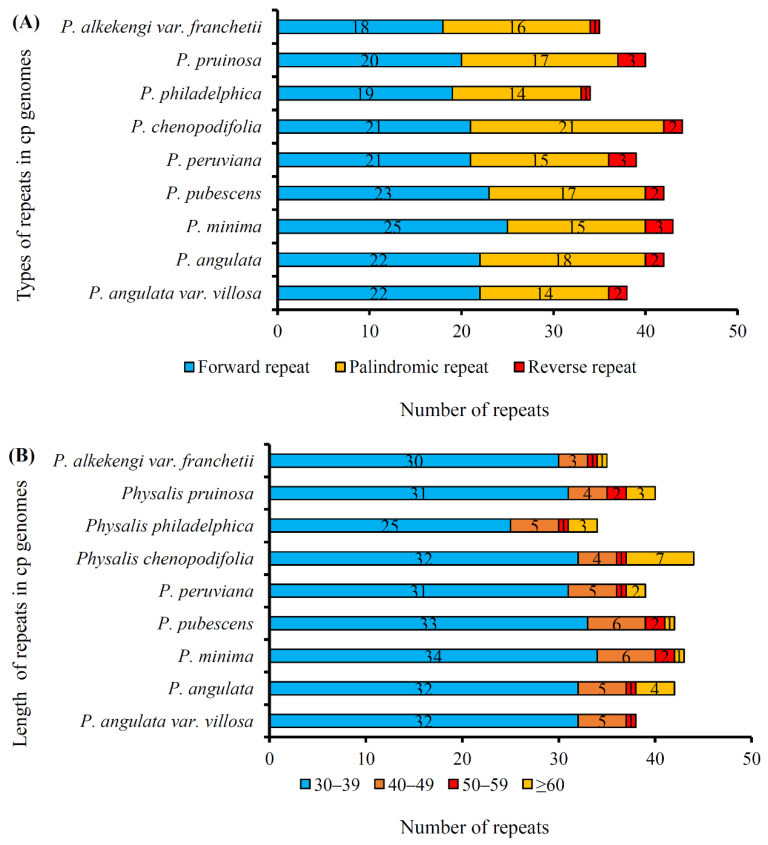
Repeated sequences in the plastomes of nine *Physalis* species. (**A**), Three repeat types (forward, reverse and palindromic repeats) in the nine *Physalis* plastomes; (**B**), Numbers of repeat (forward, reverse and palindromic) sequences by length.

**Figure 5 genes-13-02291-f005:**
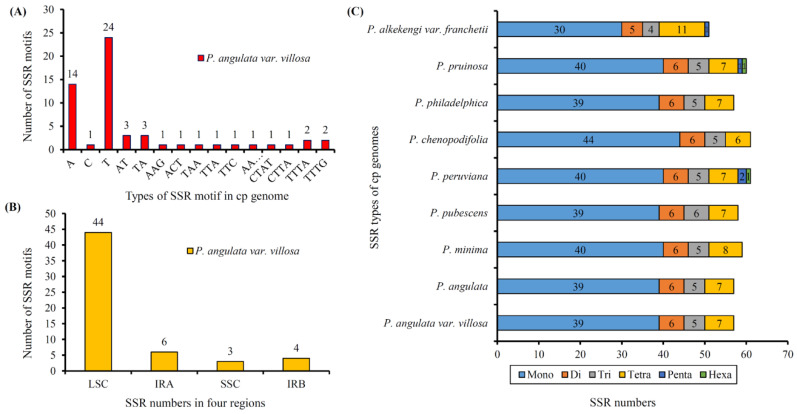
Simple sequence repeats (SSRs) types, distribution and presence in *P. angulata* var. *villosa* and other representative species from *Physalis*. (**A**), Numbers of different SSR motifs in different repeat types detected in the *P. angulata* var. *villosa* plastome. (**B**), Numbers of SSRs in different regions (IRA, IRB, LSC, and SSC) of the *P. angulata* var. *villosa* plastome. (**C**), Numbers of different SSR types detected in the genomes of the nine *Physalis* species.

**Figure 6 genes-13-02291-f006:**
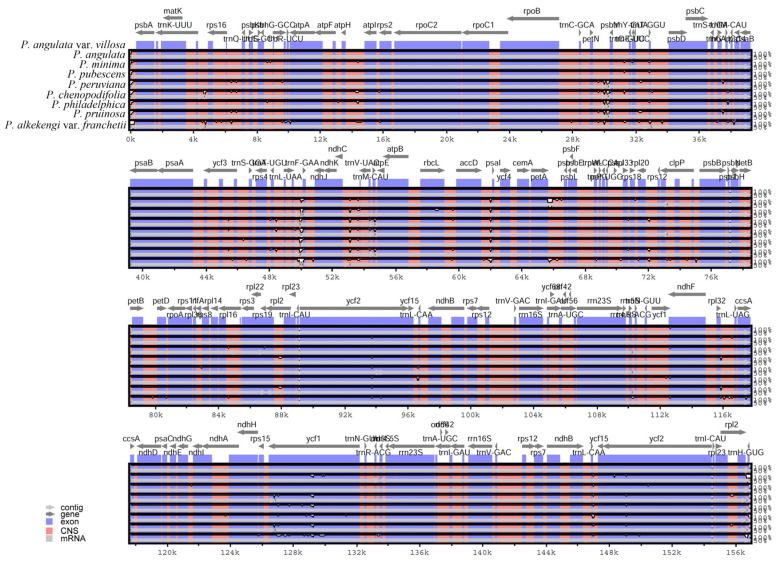
Sequence alignment of nine plastomes in the genus *Physalis* performed with mVISTA, using annotation of *P. angulata* var. *villosa* as reference. The *y*-axis presents the percentage identity, within 50–100%. Protein-encoding regions are indicated in blue and non-coding regions in red. A reduction in sequence identity is indicated by a reduction in the blue/red shadowing (white spaces).

**Figure 7 genes-13-02291-f007:**
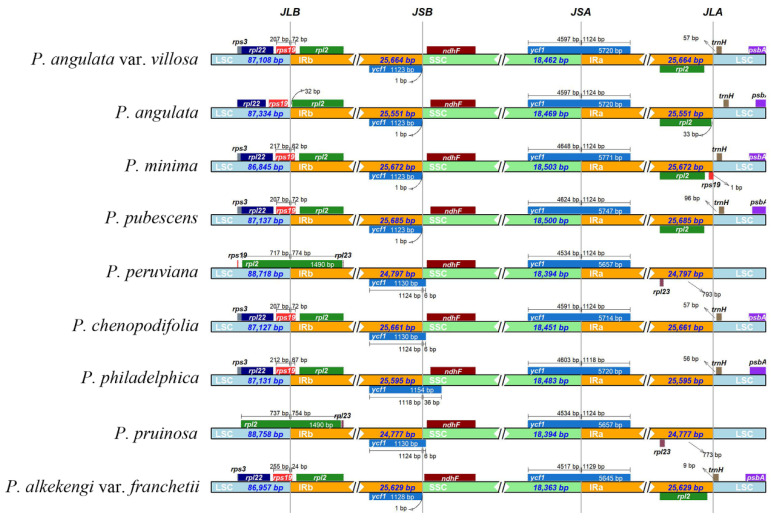
Comparison of the borders of the IR, SSC and LSC regions among nine *Physalis* plastomes. Numbers above indicate the distances in bp between the ends of the genes and the border sites.

**Figure 8 genes-13-02291-f008:**
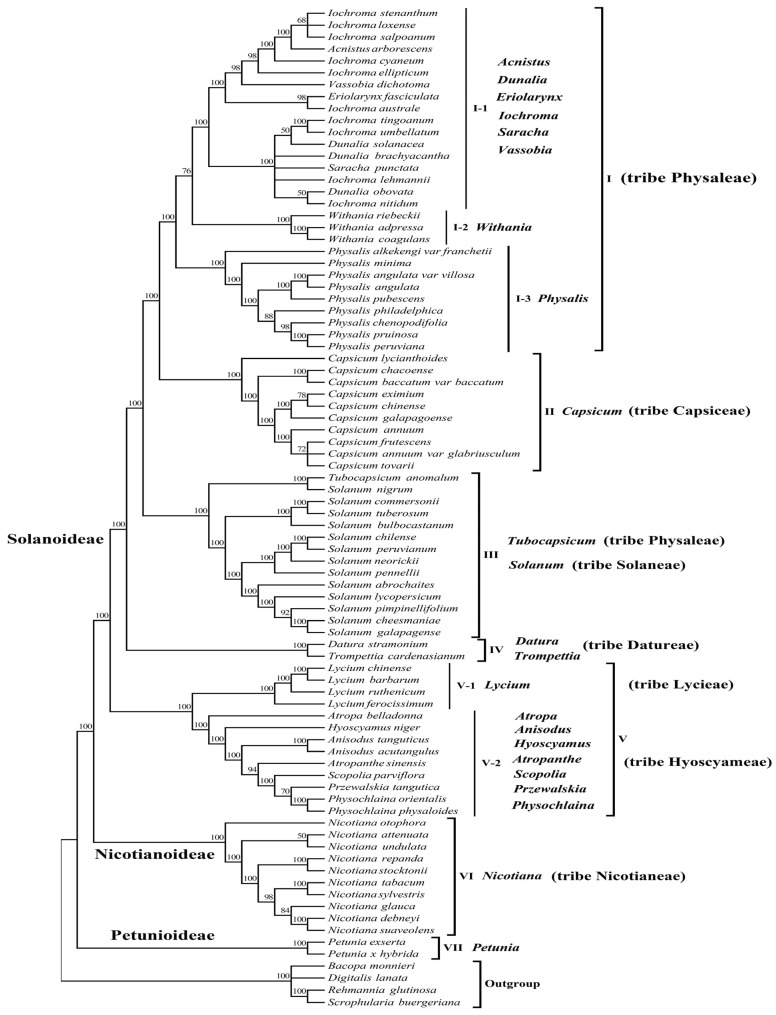
Maximum likelihood (ML) phylogenetic tree of the complete plastomes of 80 species of seven tribes (Physaleae, Capsiceae, Solaneae, Datureae, Lycieae, Hyoscyameae and Nicotianeae) from three subfamily (Solanoideae, Nicotianoideae and Petunioideae) of Solanaceae. Numbers above branches indicate bootstrap support levels.

**Table 1 genes-13-02291-t001:** Summaries of complete plastome of *P. angulata* var. *villosa*.

	*P. angulata* var. *villosa*
Genome size (bp)	156,898
LSC (bp)	87,108
SSC (bp)	18,462
IR (bp)	25,664
GC content (%)	
Total genome	37.52
LSC	35.58
SSC	31.33
IR	43.05
Gene	131
protein-coding genes	85
rRNA	8
tRNA	38
GenBank accession	OM257167

**Table 2 genes-13-02291-t002:** Genes in the plastome of *P. angulata* var. *villosa*.

Category for Genes	Group of Genes	Name of Genes
Self-replication	rRNA Genes	*rrn4.5S* (×2), *rrn5S* (×2), *rrn16S* (×2), *rrn23S* (×2)
	tRNA Genes	^a^*trnA-UGC* (×2), *trnC-GCA*, *trnD-GUC*, *trnE-UUC*, *trnF-GAA*, *trnfM-CAU*, ^a^ *trnG-GCC*, ^a^ *trnG-UCC*, *trnH-GUG*, ^a^ *trnI-GAU* (×2), *trnI-CAU*( ×2), ^a^ *trnK-UUU*, *trnL-CAA*, ^a^ *trnL-UAA*, *trnL-UAG*, *trnM-CAU*, *trnN-GUU* (×2), *trnP-UGG*, *trnQ-UUG*, *trnR-ACG* (×2), *trnR-UCU*, *trnS-GCU*, *trnS-GGA*, *trnS-UGA*, *trnT-GGU* (×2), *trnT-UGU*, *trnV-GAC* (×2), ^a^ *trnV-UAC*, *trnW-CCA*, *trnY-GUA*
	DNA dependent RNA polymerase	*rpoA*, *rpoB*, ^a^*rpoC1*, *rpoC2*
	Small subunit of ribosome	*rps2*, *rps3*, *rps4*, *rps7* (×2), *rps8*, *rps11*, ^b^ *rps12*, *rps14*, *rps15*, ^a^ *rps16*, *rps18*, *rps19*
	Large subunit of ribosome	^a^*rpl2* (×2), *rpl14*, ^a^ *rpl16*, *rpl20*, *rpl22*, *rpl23* (×2), *rpl32*, *rpl33*, *rpl36*
Photosynthesis	Subunits of ATP synthase	*atpA*, *atpB*, *atpE*, ^a^*atpF*, *atpH*, *atpI*
	Subunits of NADH-dehydrogenase	^a^*ndhA*, ^a^*ndhB* (×2), *ndhC*, *ndhD*, *ndhE*, *ndhF*, *ndhG*, *ndhH*, *ndhI*, *ndhJ*, *ndhK*
	Subunits of cytochrome b/f complex	*petA*, ^a^*petB*, ^a^*petD*, *petG*, *petL*, *petN*
	Subunits of photosystem I	*psaA*, *psaB*, *psaC*, *psaI*, *psaJ*, *ycf4*, ^b^*ycf3*
	Subunits of photosystem II	*psbA*, *psbB*, *psbC*, *psbD*, *psbE*, *psbF*, *psbH*, *psbI*, *psbJ*, *psbK*, *psbL*, *psbM*, *psbN*, *psbT, lhbA*
	Subunit of rubisco	*rbcL*
	Subunit of Acetyl-CoA-carboxylase	*accD*
	c-type cytochrom synthesis gene	*ccsA*
	Envelop membrane protein	*cemA*
	Protease	^b^ *clpP*
	Translational initiation factor	*infA*
	Maturase	*matK*
Unknown function	Conserved open reading frames	*ycf1*, *ycf2* (×2), *ycf15* (×2)

**Note:** (×2): Two gene copies in IRs; ^a^: gene containing a single intron; ^b^: gene containing two introns.

**Table 3 genes-13-02291-t003:** Genes with intron in the *P. angulata* var. *villosa* plastome and length of exons and intron.

Gene	Location	Exon I (bp)	Intron I (bp)	Exon II (bp)	Intron II (bp)	Exon III (bp)
*atpF*	LSC	145	710	410		
*clpP*	LSC	71	789	292	640	228
*ndhA*	SSC	553	1147	539		
*ndhB*	IRB	777	679	756		
*ndhB*	IRA	777	679	756		
*petB*	LSC	6	753	642		
*petD*	LSC	8	759	475		
*rpl2*	IRA	391	666	434		
*rpl2*	IRB	391	666	434		
*rpl16*	LSC	9	959	396		
*rpoC1*	LSC	432	737	1614		
*rps12* ^①^	LSC+IRA	234	-	25	536	114
*rps12* ^②^	LSC+IRB	232	-	26	536	114
*rps16*	LSC	40	855	227		
*ycf3*	LSC	124	736	230	782	153
*trnA-UGC*	IRA	37	811	36		
*trnA-UGC*	IRB	36	811	37		
*trnE-UUC*	IRA	32	724	40		
*trnE-UUC*	IRB	40	724	32		
*trnI-GAU*	IRA	36	16	36		
*trnI-GAU*	IRB	36	16	36		
*trnK-UUU*	LSC	36	2509	37		
*trnL-UAA*	LSC	35	497	50		
*trnV-UAC*	IRB	56	553	36		
*trnS-CGA*	LSC	31	676	60		

**Note:** Hyphen-: spliceosomal intron; ①, ②: The rps12 gene is divided into 5′-rps12 in LSC region, ① 3′-rps12 in IRA region and ② 3′-rps12 in IRB region.

**Table 4 genes-13-02291-t004:** Codon—anticodon recognition patterns and codon usage of the *P. angulata* var. *villosa* plastome.

Codon.	Amino Acid	Count	RSCU	tRNA	Codon	Amino Acid	Count	RSCU	tRNA
UUU	Phe	2268.0	1.22		UAU	Tyr	1433.0	1.34	
UUC	Phe	1453.0	0.78	*trnF-GAA*	UAC	Tyr	708.0	0.66	*trnY-GUA*
UUA	Leu	1087.0	1.27	*trnL-UAA*	UAA	TER	1256.0	1.21	
UUG	Leu	1080.0	1.26	*trnL-CAA*	UAG	TER	784.0	0.75	
CUU	Leu	1107.0	1.29		CAU	His	855.0	1.38	
CUC	Leu	673.0	0.78		CAC	His	387.0	0.62	*trnH-GUG*
CUA	Leu	731.0	0.85	*trnL-UAG*	CAA	Gln	1049.0	1.43	*trnQ-UUG*
CUG	Leu	466.0	0.54		CAG	Gln	423.0	0.57	
AUU	Ile	1855.0	1.27	*trnI-CAU*	AAU	Asn	1774.0	1.42	
AUC	Ile	1127.0	0.77	*trnI-GAU*	AAC	Asn	722.0	0.58	*trnN-GUU*
AUA	Ile	1417.0	0.97		AAA	Lys	2115.0	1.36	*trnK-UUU*
AUG	Met	868.0	1.00	*trnM-CAU*	AAG	Lys	1004.0	0.64	
GUU	Val	817.0	1.40		GAU	Asp	1059.0	1.44	
GUC	Val	460.0	0.79	*trnV-GAC*	GAC	Asp	413.0	0.56	*trnD-GUC*
GUA	Val	668.0	1.14	*trnV-UAC*	GAA	Glu	1399.0	1.38	*trnE-UUC*
GUG	Val	392.0	0.67		GAG	Glu	628.0	0.62	
UCU	Ser	1168.0	1.47	*trnS-GCU*	UGU	Cys	713.0	1.20	
UCC	Ser	911.0	1.15	*trnS-GGA*	UGC	Cys	473.0	0.80	*trnC-GCA*
UCA	Ser	923.0	1.16	*trnS-UGA*	UGA	TER	1086.0	1.04	
UCG	Ser	583.0	0.73		UGG	Trp	665.0	1.00	*trnW-CCA*
CCU	Pro	677.0	1.09	*trnP-UGG*	CGU	Arg	413.0	0.73	*trnR-ACG*
CCC	Pro	646.0	1.04		CGC	Arg	230.0	0.41	*trnR-UCU*
CCA	Pro	757.0	1.22		CGA	Arg	645.0	1.15	
CCG	Pro	400.0	0.65		CGG	Arg	373.0	0.66	
ACU	Thr	679.0	1.14		AGU	Arg	682.0	0.86	
ACC	Thr	614.0	1.03	*trnT-GGU*	AGC	Arg	499.0	0.63	
ACA	Thr	717.0	1.21	*trnT-UGU*	AGA	Arg	1091.0	1.94	
ACG	Thr	364.0	0.61		AGG	Arg	625.0	1.11	
GCU	Ala	500.0	1.25		GGU	Gly	594.0	1.04	
GCC	Ala	389.0	0.97		GGC	Gly	349.0	0.61	*trnG-GCC*
GCA	Ala	471.0	1.18	*trnA-UGC*	GGA	Gly	800.0	1.40	*trnG-UCC*
GCG	Ala	241.0	0.60		GGG	Gly	543.0	0.95	

**Table 5 genes-13-02291-t005:** Repeat sequences present in the *P. angulata* var. *villosa* plastome.

Number	Repeat Size	Repeat Type	Repeat Position 1	Repeat Location 1	Repeat Position 2	Repeat Location 2	E-Value
1	56	P	79,608	IGS (*petB*-*petD*)	79,608	IGS (*petB*-*petD*)	1.85 × 10^−20^
2	48	P	77,177	*psbT*	77,177	*psbT*	8.74 × 10^−20^
3	39	F	40,031	*psaB*	42,255	*psaA*	2.29 × 10^−14^
4	48	F	50,038	IGS *(trnL-UAA*-*trnF-GAA*)	50,088	IGS (*trnL-UAA*-*trnF-GAA*)	4.08 × 10^−14^
5	41	F	101,181	IGS (rps7-trnV-GAC)	122,830	*ndhA*	1.76 × 10^−13^
6	41	P	122,830	*ndhA*	142,400	*ndhA*	1.76 × 10^−13^
7	37	F	53,125	IGS (*ndhC*-*trnV-UAC*)	53,145	IGS (*ndhC*-*trnV-UAC*)	3.67 × 10^−13^
8	39	F	45,095	*ycf3*	122,832	*ndhA*	2.68 × 10^−12^
9	40	F	50,000	IGS (*trnL-UAA*-*trnF-GAA*)	50,100	IGS (*trnL-UAA*-*trnF-GAA*)	4.02 × 10^−11^
10	39	F	45,095	*ycf3*	101,183	IGS (*rps7*-*trnV-GAC*)	1.53 × 10^−10^
11	39	P	45,095	*ycf3*	142,400	IGS (*trnV-GAC*-*rps7*)	1.53 × 10^−10^
12	35	P	96,669	IGS (*ycf15*-*trnL-CAA*)	96,669	IGS (*ycf15*-*trnL-CAA*)	6.16 × 10^−10^
13	35	F	96,669	IGS (*ycf15*-*trnL-CAA*)	146,918	IGS *(trnL-CAA*-*ycf15*)	6.16 × 10^−10^
14	35	P	146,918	IGS *(trnL-CAA*-*ycf15*)	146,918	IGS *(trnL-CAA*-*ycf15*)	6.16 × 10^−10^
15	30	P	8,301	IGS *(psbI*-*trnS-GCU*)	46,827	*trnS-GGA*	6.01 × 10^−9^
16	36	F	28,430	*trnC-GCA*	49,976	IGS (*trnL-UAA*-*trnF-GAA*)	8.31 × 10^−9^
17	36	R	129,718	*ycf1*	129,718	*ycf1*	8.31 × 10^−9^
18	33	F	38,168	IGS (*trnfM-CAU*-*rps14*)	38,197	IGS (*trnfM-CAU*-*rps14*)	9.29 × 10^−9^
19	37	F	94,191	*ycf2*	94,209	*ycf2*	7.69 × 10^−8^
20	37	P	94,191	*ycf2*	149,376	*ycf2*	7.69 × 10^−8^
21	37	P	94,209	*ycf2*	149,394	*ycf2*	7.69 × 10^−8^
22	37	F	149,376	*ycf2*	149,394	*ycf2*	7.69 × 10^−8^
23	34	F	109,819	IGS (*rrn4.5S*-*rrn5S*)	109,851	IGS (*rrn4.5S*-*rrn5S*)	1.18 × 10^−7^
24	34	P	109,819	IGS (*rrn4.5S*-*rrn5S*)	133,737	IGS (*rrn5S*-*rrn4.5S*)	1.18 × 10^−7^
25	34	P	109,851	IGS (*rrn4.5S*-*rrn5S*)	133,769	IGS (*rrn5S*-*rrn4.5S*)	1.18 × 10^−7^
26	34	F	133,737	IGS (*rrn5S*-*rrn4.5S*)	133,769	IGS (*rrn5S*-*rrn4.5S*)	1.18 × 10^−7^
27	36	F	50,000	IGS (*trnL-UAA*-*trnF-GAA*)	50,050	IGS (*trnL-UAA*-*trnF-GAA*)	2.83 × 10^−7^
28	33	F	50,113	IGS (*trnL-UAA*-*trnF-GAA*)	50,193	*trnF-GAA*	1.38 × 10^−5^
29	31	F	8,300	IGS (*psbI*-*trnS-GCU*)	36,776	*trnS-UGA*	1.82 × 10^−4^
30	31	F	9,743	*trnG-GCC*	37,739	*trnG-UCC*	1.82 × 10^−4^
31	30	F	36,630	IGS (*psbC*-*trnS-UGA*)	74,420	*clpP*	6.58 × 10^−4^
32	30	P	36,777	IGS (*psbC*-*trnS-UGA*)	46,827	*trnS-GGA*	6.58 × 10^−4^
33	30	F	45,107	*ycf3*	101,195	IGS (*rps7*-*trnV-GAC*)	6.58 × 10^−4^
34	30	P	45,107	*ycf3*	142,397	IGS (trnV-GAC-rps7)	6.58 × 10^−4^
35	30	F	101,195	IGS (*rps7*-*trnV-GAC*)	122,844	IGS (*ndhI*-*ndhA*)	6.58 × 10^−4^
36	30	P	122,844	IGS (*ndhI*-*ndhA*)	142,397	*ndhA*	6.58 × 10^−4^
37	30	R	129,723	*ycf1*	129,731	*ycf1*	6.58 × 10^−4^
38	30	F	149,384	*ycf2*	149,402	*ycf2*	6.58 × 10^−4^

## Data Availability

All data are included in the manuscript, and the complete plastome of *Physalis angulata* var. *villosa* were submitted to the NCBI database (https://www.ncbi.nlm.nih.gov/ (accessed on 13 January 2022)) with GenBank accession numbers OM257167.

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
