# Peer review of "Complete Plastome of Physalis angulata var. villosa, Gene Organization, Comparative Genomics and Phylogenetic Relationships among Solanaceae"

_genes, 2022, doi:10.3390/genes13122291_

Round 1

Reviewer 1 Report

the paper is well-written and well structured 

the part of introduction needs to explain the importance of the phylogenetic studies for this family or genus before. 

The part of the discussion needs to explain more the results and compare them with previous studies more accurately because most of the discussion reviews the results only.

Author Response

Response to Reviewer#1

Dear Editors and Reviewers:

Thank you for your letter and for the reviewers’ comments concerning our manuscript entitled Complete plastome of Physalis angulata var. villosa, gene organization, comparative genomics and phylogenetic relationships among Solanaceae” (Manuscript ID: genes-2008876). Those comments are all valuable and very helpful for revising and improving our paper, as well as the important guiding significance to our researches. We have studied comments carefully and have made correction which we hope meet with approval. Revised portion are marked in red in the paper. The main corrections in the paper and the responds to the reviewer’s comments are as flowing:

Responds to the reviewer #1’s comments:

Comments and Suggestions for Authors

  1. the paper is well-written and well structured

Response:Thank you for your comments.

  1. the part of introduction needs to explain the importance of the phylogenetic studies for this family or genus before.

Response:The reviewer's suggestions are greatly appreciated. We have added some content in the introduction section, hoping to meet your requirements. (Line 71-92)

  1. The part of the discussion needs to explain more the results and compare them with previous studies more accurately because most of the discussion reviews the results only.

Response:Considering the Reviewer’s suggestion, we have added some content in the Discussion section. Thank you so much! (Discussion section)

We tried our best to improve the MS and made some changes in the MS. These changes will not influence the content and framework of the paper. And here we did not list the changes but marked in red in revised paper.

We appreciate for Editors/Reviewers’ warm work earnestly, and hope that the correction will meet with approval.

Once again, thank you very much for your comments and suggestions.

Yours sincerely,

Shangguo Feng

College of Life and Environmental Sciences, Hangzhou Normal University, Hangzhou 311121, China.

Reviewer 2 Report

This study by Xiaori Zhan et al. presents plastid genome of Physalis angulata var. villosa (Solanaceae family) and results of comparative analysis of available Physalis plastomes; they also performed phylogenetic analyses of Solanaceae plastomes. In the newly generated complete plastome sequence authors identified variable regions, which can be useful in plant barcoding, and molecular phylogenetic anaysis showed relationships among 80 Solanaceae plastomes. The results are interesting; however there are several points throughout the manuscript which need to be addressed.

 In Introduction, some words should be said about Physalis angulata and its similarity/difference with P.angulata var. villosa. This will improve justification of P.angulata var. villosa plastome sequencing, since the plastome of Physalis angulata is freely accessible in GenBank. Further, explicit comparison of their plastomes along with other Physalis plastomes should be performed and discussed.

 As an examination of phylogenetic relationships within Solanaceae is one of purposes of the presented study, some information about current view on taxonomy and phylogeny of the family will be also helpful if provided in Introduction.

 Results, section 3.1: authors counted rps12 as a duplicated gene, but it is not, because a 5'-part of this trans-spliced gene is located in the LSC region. Otherwise, ycf1 also can be considered duplicated.

 Results, in Table 2: lhbA (encoding light harvesting antenna) is a synonym to psbZ (photosystem II reaction center protein Z) and should be listed among subunits of photosystem II (see B.R. Green and W.W. Parson, Light-Harvesting Antennas in Photosynthesis), whereas ycf3 is a photosystem I assembly protein (see Naver H, Boudreau E, Rochaix J, 2001, Functional studies of Ycf3: Its role in assembly of photosystem I and interactions with some of its subunits. Plant Cell 13:2731–2745) and should not be in a category of genes with unknown function.

 In this study, the RNA editing sites were predicted by the only software PREP, neither other programs have been tried nor cDNA sequencing performed for confirmation. However, not all predicted sites are actually edited (see, e.g., doi: 10.1080/17429145.2022.2101700). Indeed, 150 editing sites is three times more than reported for other Solanaceae, therefore these results should be interpreted and discussed with greatest caution (using "probably, presumably, potentially, perhaps" etc). The table 5 is huge and would be better placed in Supplement, to my mind.

 Results, section 3.4, Figure 6: IR, LSC and SSC borders in some Physalis plastomes defined and presented incorrectly. First of all, the first inverted repeat is designated as IRb, the second is IRa, so the right order is LSC-IRb-SSC-IRa (see Figure 1 also). Further, my quick examination of available in GenBank data showed, that in all but P.priunosa plastomes the LSC/IRb border crosses rps19 and the first gene in IRb (and the last gene in IRa) is rpl2. Therefore, this part of analysis should be repeated, and Results and Discussion should be corrected accordingly. It should be noted, that accession MH045575 (Alkekengi plastome) contains wrong plastome start point and this should be corrected prior the analysis, as well as checking of other plastomes annotation with correction made if needed.

 Results of phylogenetic analysis are doubtful for me for the following reason. In section 3.2 authors reported presence of inverted repeats both in non-coding and coding sequences; no doubt they are present in other Solanaceae plastomes also. However, it is well known that inverted repeats may be associated with inversions, which may occur independently in different lineages. In phylogenetic analysis such inversions may affect the inferred topology and branch length estimation, as well as branch support assessment (see, e.g., doi: 10.1002/ajb2.1775); and therefore it is wise to remove them or reverse complement prior phylogenetic analysis (e.g., doi: 10.3390/plants11050709). How did authors handle this possible source of homoplasy in their plastome data? If the data were analyzed "as such", then the results may be skewed.

 Besides, I make few comments as follows:

I recommend recent interesting papers on Solanaceae plastomes (https://doi.org/10.1016/j.ccmp.2021.100002 and https://doi.org/10.3897/phytokeys.210.85668) for comparison and discussion of presented results

Check Table S1, where is P.alkekengi? Also, in some publications, Alkekengi officinarum is synonymized with P.alkekengi, clarify this point.

Line 21: "It was circular" - plastid genome conformations vary greatly, not only circles, but also linear, concatenated and branched multimers.

Line 29: less highly concerved - less or highly?

Line 58: "rate is relatively slow [6,9]" - papers of 2015 and 2020 referenced here, while this feature is well known since publications by Wolfe et al (1987, PNAS 84:9054-9058) and Palmer (1991, Plastid chromosomes: structure and evolution)!

Line 80: "laid a foundation" - definitely, such a statement is premature, albeit results obtained here may be useful for "the development of..." indeed

Line 131: MAFFT is used for alignment, it does not produce any analysis.

Line 135: Large deletions are common in plastid sequences, so "complete gap elimination" may greatly reduce an amount of useful data in alignment. Elimination of half-gapped columns is more common

Line 263-264: does this mean, that other genes do not show divergence and they are identical? If yes, I do not believe that.

Line 274 "some level of expansion"? Define and explain levels 1, 2 etc.

Figure 6 produces a confusing illusion that all presented genes are transcribed in one direction. Compare with Figure 4 in Huang et al (doi: 10.1016/j.ccmp.2021.100002)

Figure 7: Indication of subfamilies and tribes here will help to see, to what extent presented plastome phylogeny corresponds to current taxonomy in Solanaceae

Line 337: pioneering and very important publications should be credited here (Ogihara et al 1988 doi: 10.1073/pnas.85.22.8573, Palmer 1991 doi: 10.1016/B978-0-12-715007-9.50009-8, Chumley et al 2006 doi: 10.1093/molbev/msl089, etc) instead of papers 2020!

Lines 386-394: Introduction is a more appropriate place for this paragraph.

Line 409 and 424: non-homologous genera? What is that?

Author Response

Response to Reviewer#2

Dear Editors and Reviewers:

Thank you for your letter and for the reviewers’ comments concerning our manuscript entitled Complete plastome of Physalis angulata var. villosa, gene organization, comparative genomics and phylogenetic relationships among Solanaceae” (Manuscript ID: genes-2008876). Those comments are all valuable and very helpful for revising and improving our paper, as well as the important guiding significance to our researches. We have studied comments carefully and have made correction which we hope meet with approval. Revised portion are marked in red in the paper. The main corrections in the paper and the responds to the reviewer’s comments are as flowing:

Responds to the reviewer #2’s comments:

This study by Xiaori Zhan et al. presents plastid genome of Physalis angulata var. villosa (Solanaceae family) and results of comparative analysis of available Physalis plastomes; they also performed phylogenetic analyses of Solanaceae plastomes. In the newly generated complete plastome sequence authors identified variable regions, which can be useful in plant barcoding, and molecular phylogenetic anaysis showed relationships among 80 Solanaceae plastomes. The results are interesting; however there are several points throughout the manuscript which need to be addressed.

  1. In Introduction, some words should be said about Physalis angulata and its similarity/difference with P.angulata var. villosa. This will improve justification of P.angulata var. villosa plastome sequencing, since the plastome of Physalis angulata is freely accessible in GenBank. Further, explicit comparison of their plastomes along with other Physalis plastomes should be performed and discussed.

Response:First of all, thank you very much for Reviewer's valuable advice. We have added some describes about differences between P.angulata var. villosa and other Physalis plants in the Introduction section. And, in order to more intuitive understanding of the P.angulata var. villosa, we added a phenotype figure of P.angulata var. villosa. In addition, considering the Reviewer’s suggestion, some discussion contents have been added in the Discussion section (Figure 1; Line 42-51; Line 367-371).

  1. As an examination of phylogenetic relationships within Solanaceae is one of purposes of the presented study, some information about current view on taxonomy and phylogeny of the family will be also helpful if provided in Introduction.

Response: Considering the Reviewer's suggestion, we have added some relevant content in the Introduction section, hoping to meet your requirements. (Line 71-92)

  1. Results, section 3.1: authors counted rps12 as a duplicated gene, but it is not, because a 5'-part of this trans-spliced gene is located in the LSC region. Otherwise, ycf1 also can be considered duplicated.

Response:Thank you for your valuable suggestions. Considering the Reviewer's suggestion, we have made the corresponding changes in Table 2 and the MS.

  1. Results, in Table 2: lhbA (encoding light harvesting antenna) is a synonym to psbZ (photosystem II reaction center protein Z) and should be listed among subunits of photosystem II (see B.R. Green and W.W. Parson, Light-Harvesting Antennas in Photosynthesis), whereas ycf3 is a photosystem I assembly protein (see Naver H, Boudreau E, Rochaix J, 2001, Functional studies of Ycf3: Its role in assembly of photosystem I and interactions with some of its subunits. Plant Cell 13:2731–2745) and should not be in a category of genes with unknown function.

Response:Thank you very much for your valuable suggestions! We have made corresponding changes in Table 2 and the MS.

  1. In this study, the RNA editing sites were predicted by the only software PREP, neither other programs have been tried nor cDNA sequencing performed for confirmation. However, not all predicted sites are actually edited (see, e.g., doi: 10.1080/17429145.2022.2101700). Indeed, 150 editing sites is three times more than reported for other Solanaceae, therefore these results should be interpreted and discussed with greatest caution (using "probably, presumably, potentially, perhaps" etc). The table 5 is huge and would be better placed in Supplement, to my mind.

Response:We have made corresponding changes in the MS and put Table 5 in the Supplement. Thank you very much! (Table S2)

  1. Results, section 3.4, Figure 6: IR, LSC and SSC borders in some Physalis plastomes defined and presented incorrectly. First of all, the first inverted repeat is designated as IRb, the second is IRa, so the right order is LSC-IRb-SSC-IRa (see Figure 1 also). Further, my quick examination of available in GenBank data showed, that in all but P.priunosa plastomes the LSC/IRb border crosses rps19 and the first gene in IRb (and the last gene in IRa) is rpl2. Therefore, this part of analysis should be repeated, and Results and Discussion should be corrected accordingly. It should be noted, that accession MH045575 (Alkekengi plastome) contains wrong plastome start point and this should be corrected prior the analysis, as well as checking of other plastomes annotation with correction made if needed.

Response:Yes, it is really true as Reviewer point out that Figure 6 does have some problems. We have done it again, hoping it could meet your requirements. Thank you very much! (Figure 7 in the revised MS)

  1. Results of phylogenetic analysis are doubtful for me for the following reason. In section 3.2 authors reported presence of inverted repeats both in non-coding and coding sequences; no doubt they are present in other Solanaceae plastomes also. However, it is well known that inverted repeats may be associated with inversions, which may occur independently in different lineages. In phylogenetic analysis such inversions may affect the inferred topology and branch length estimation, as well as branch support assessment (see, e.g., doi: 10.1002/ajb2.1775); and therefore it is wise to remove them or reverse complement prior phylogenetic analysis (e.g., doi: 10.3390/plants11050709). How did authors handle this possible source of homoplasy in their plastome data? If the data were analyzed "as such", then the results may be skewed.

Response:Thanks so much for the Reviewer’s valuable suggestions. We used the method of phylogenetic tree construction based on the whole chloroplast genome, which is also the method adopted in most related research papers. In addition, our results were compared with other published papers and found to be in good confidence. At the same time, considering the reviewer's suggestion, we redrew the phylogenetic tree and found that our clustering was also relatively consistent with the traditional classification. Of course, we will definitely study the suggestions put forward by the reviewers, and believe that they will be of great help to our future research. Thank you again for your valuable advice.

 Besides, I make few comments as follows:

  1. I recommend recent interesting papers on Solanaceae plastomes (https://doi.org/10.1016/j.ccmp.2021.100002 and https://doi.org/10.3897/phytokeys.210.85668) for comparison and discussion of presented results

Response:Thank you very much for the papers provided by the Reviewer, which are new and interesting, and are very helpful for our further and more comprehensive study on the phylogenetic evolution of Solanaceae. In another study we are carrying out, we will cite the results of these two literatures and make in-depth comparative analysis.

  1. Check Table S1, where is P.alkekengi? Also, in some publications, Alkekengi officinarum is synonymized with P.alkekengi, clarify this point.

Response:I'm terribly sorry for our negligence. we have made corresponding changes in the Table S1 and the MS. (Table S1; Line 241)

  1. Line 21: "It was circular" - plastid genome conformations vary greatly, not only circles, but also linear, concatenated and branched multimers.

Response: Thank you for your suggestion, we have made the corresponding changes n in the MS.

  1. Line 29: less highly concerved - less or highly?

Response: I am very sorry that our language is not accurate, and we have made corresponding changes.

  1. Line 58: "rate is relatively slow [6,9]" - papers of 2015 and 2020 referenced here, while this feature is well known since publications by Wolfe et al (1987, PNAS 84:9054-9058) and Palmer (1991, Plastid chromosomes: structure and evolution)!

Response: Thank you very much! We have made corresponding changes in the MS. (Line 65)

  1. Line 80: "laid a foundation" - definitely, such a statement is premature, albeit results obtained here may be useful for "the development of..." indeed

Response: I am very sorry that our language is not accurate, and we have made corresponding modifications in the MS. (Line 98)

  1. Line 131: MAFFT is used for alignment, it does not produce any analysis.

Response: We have made corresponding changes in the MS. Thank you so much! (Line 149)

  1. Line 135: Large deletions are common in plastid sequences, so "complete gap elimination" may greatly reduce an amount of useful data in alignment. Elimination of half-gapped columns is more common

Response: Thank you very much for your valuable suggestions, which will have important guiding significance for our future researches.

  1. Line 263-264: does this mean, that other genes do not show divergence and they are identical? If yes, I do not believe that.

Response: I am sorry that our statement has caused you to misunderstand. We have made corresponding changes. ‘In addition to that, protein-coding genes, such as ycf1, ycf2, ndhF, rps19 and ccsA, also showed high sequence variation’. (Line 284)

  1. Line 274 "some level of expansion"? Define and explain levels 1, 2 etc.

Response: We have made corresponding changes in the MS. Thank you so much! The expansion and contraction were considered to be the evolutionary events and the main cause of plastome size variation. (Line 290-296)

  1. Figure 6 produces a confusing illusion that all presented genes are transcribed in one direction. Compare with Figure 4 in Huang et al (doi: 10.1016/j.ccmp.2021.100002)

Response: We have redrawn Figure 6, hoping to meet your requirements. Thank you very much! (Figure 7 in the revised MS)

  1. Figure 7: Indication of subfamilies and tribes here will help to see, to what extent presented plastome phylogeny corresponds to current taxonomy in Solanaceae

Response: Thank you very much for your valuable suggestions. We have redrawn Figure 7 and Figure S1, and made corresponding changes in Table S1 and the MS. hoping to meet your requirements. (Figure 8, Figure S1 and Table S1 in the revised MS)

  1. Line 337: pioneering and very important publications should be credited here (Ogihara et al 1988 doi: 10.1073/pnas.85.22.8573, Palmer 1991 doi: 10.1016/B978-0-12-715007-9.50009-8, Chumley et al 2006 doi: 10.1093/molbev/msl089, etc) instead of papers 2020!

Response: We have made the corresponding changes in the MS. Thank you very much for your careful review. Your suggestions are very helpful to the improvement of our paper. (Line 362)

  1. Lines 386-394: Introduction is a more appropriate place for this paragraph.

Response: Considering the Reviewer’s suggestion, we have made corresponding changes in the MS. (Line 81-92)

  1. Line 409 and 424: non-homologous genera? What is that?

 Response: I am sorry that our statement has caused you to misunderstand. We have made corresponding changes in the MS. (Line 427 and Line 449)

We tried our best to improve the MS and made some changes in the MS. These changes will not influence the content and framework of the paper. And here we did not list the changes but marked in red in revised paper.

We appreciate for Editors/Reviewers’ warm work earnestly, and hope that the correction will meet with approval.

Once again, thank you very much for your comments and suggestions.

Yours sincerely,

Shangguo Feng

College of Life and Environmental Sciences, Hangzhou Normal University, Hangzhou 311121, China.

Reviewer 3 Report

 The title is well-appropriate having good academic significance.

The topics addressed has good scientific depth.

However abstract is concise and will be summarized.

Results of the research work are well. If possible properly explain table and figures caption. 

If possible the phylogenetic tree resolution should be improve.

Author Response

Response to Reviewer#3

 Dear Editors and Reviewers:

Thank you for your letter and for the reviewers’ comments concerning our manuscript entitled Complete plastome of Physalis angulata var. villosa, gene organization, comparative genomics and phylogenetic relationships among Solanaceae” (Manuscript ID: genes-2008876). Those comments are all valuable and very helpful for revising and improving our paper, as well as the important guiding significance to our researches. We have studied comments carefully and have made correction which we hope meet with approval. Revised portion are marked in red in the paper. The main corrections in the paper and the responds to the reviewer’s comments are as flowing:

Responds to the reviewer #3’s comments:

1.The title is well-appropriate having good academic significance.

 Response: Thank you for your comments.

  1. The topics addressed has good scientific depth.

 Response: Thank you for your comments.

  1. However abstract is concise and will be summarized.

 Response: Considering the reviewer's suggestion, we have made some changes in the abstract section.

  1. Results of the research work are well. If possible properly explain table and figures caption. 

Response: As suggested by the reviewer, we have added some explanations to the titles of some tables and figures.

  1. If possible the phylogenetic tree resolution should be improve.

Response: Considering the reviewer's suggestion, we have redrawn the picture of the phylogenetic tree (Figure 8 and Figure S1 in the revised MS).

We tried our best to improve the MS and made some changes in the MS. These changes will not influence the content and framework of the paper. And here we did not list the changes but marked in red in revised paper.

We appreciate for Editors/Reviewers’ warm work earnestly, and hope that the correction will meet with approval.

Once again, thank you very much for your comments and suggestions.

Yours sincerely,

Shangguo Feng

College of Life and Environmental Sciences, Hangzhou Normal University, Hangzhou 311121, China.

Round 2

Reviewer 2 Report

Authors have adequately addressed my most important comments and have made many improvements.

But I would like to draw authors' attention to Figure 7 and text (lines 297-304 and 392-399) which show and describe differences which actually do not exist. I examined Physalis plastome data from GenBank again and found them extremely similar in LSC/IR junctions. Plastome's compartments are not identical in length, but contain the same set of genes at IR junctions in all examined Physalis plastomes; and the Figure 7 should reflect this similarity. Strictly speaking, only Ph.minima is presented correctly here, all others require correction to match with the picture of Ph.minima. Authors should realize that software like IRscope (or similar) presents picture according to provided annotation; and when the annotation is incomplete, then the picture is wrong. Genbank accession MN508249 (Physalis chenopodifolia plastome) ends with rpl23 annotation, but this does not mean, that it's IRa does not contain rpl2 gene and rps19-fragment, it just means that not all genes are annotated scrupulously. The same is true for other plastomes, as I wrote earlier, authors should have checked all Physalis plastomes annotation with correction made if needed. In the present form, Figure 7 is misleading, as well as text in lines lines 297-304 and 392-399.

One additional note - in Table S1 Solanum abrochaites should be Solanum habrochaites.

Author Response

Response to Editors and Reviewer#2 (Round 2)

Dear Editors and Reviewer:

Once again, thank you for your letter and for the reviewer’s comments concerning our manuscript entitled “Complete plastome of Physalis angulata var. villosa, gene organization, comparative genomics and phylogenetic relationships among Solanaceae” (Manuscript ID: genes-2008876). Those comments are all valuable and very helpful for revising and improving our paper, as well as the important guiding significance to our researches. We have studied comments carefully and have made correction again which we hope meet with approval. Revised portion are marked in red in the paper. The main corrections in the paper and the responds to the reviewer’s comments are as flowing:

Responds to the reviewer #2’s comments:

  1. Authors have adequately addressed my most important comments and have made many improvements.

Response: We are very grateful for your valuable suggestions, which are of great help to the improvement of our manuscript.

  1. But I would like to draw authors' attention to Figure 7 and text (lines 297-304 and 392-399) which show and describe differences which actually do not exist. I examined Physalisplastome data from GenBank again and found them extremely similar in LSC/IR junctions. Plastome's compartments are not identical in length, but contain the same set of genes at IR junctions in all examined Physalisplastomes; and the Figure 7 should reflect this similarity. Strictly speaking, only Ph.minima is presented correctly here, all others require correction to match with the picture of Ph.minima. Authors should realize that software like IRscope (or similar) presents picture according to provided annotation; and when the annotation is incomplete, then the picture is wrong. Genbank accession MN508249 (Physalis chenopodifolia plastome) ends with rpl23 annotation, but this does not mean, that it's IRa does not contain rpl2 gene and rps19-fragment, it just means that not all genes are annotated scrupulously. The same is true for other plastomes, as I wrote earlier, authors should have checked all Physalis plastomes annotation with correction made if needed. In the present form, Figure 7 is misleading, as well as text in lines lines 297-304 and 392-399.

Response: Thank you very much for your valuable advice. According to the reviewer's suggestion, we carefully checked the information of sequences and redrew Figure 7 using the R script of IRscope software and made some corresponding modifications in the MS, hoping to meet your requirements. Most of the sequences were downloaded from the GenBank database, and these data have been reported, so we refer to many relevant studies and took the default annotation approach. We are especially grateful to the reviewer for giving us so much help and guidance, which will be of great help to our future research.

  1. One additional note - in Table S1 Solanum abrochaitesshould be Solanum habrochaites.

Response: I am very sorry for the writing error caused by our negligence, and we have corrected it. Your serious, rigorous, meticulous attitude makes us admire. Thank you very much again.

We tried our best to improve the MS and made some changes in the MS. These changes will not influence the content and framework of the paper. And here we did not list the changes but marked in red in revised paper.

We appreciate for Editors/Reviewers’ warm work earnestly, and hope that the correction will meet with approval.

Once again, thank you very much for your comments and suggestions.

Yours sincerely,

Shangguo Feng

College of Life and Environmental Sciences, Hangzhou Normal University, Hangzhou 311121, China.
